# Connecting the Smart Village: A Switch towards Smart and Sustainable Rural-Urban Linkages in Spain

Cristina García Fernández * and Daniël Peek

Public Economics and Political Economy, Faculty of Political Sciences and Sociology, Universidad Complutense de Madrid (UCM), 28223 Madrid, Spain
*   Correspondence: cgarciaf@ucm.es

**Abstract:** This research focuses on the dimensions of the Smart Village concept to understand to what extent smart development in the countryside could contribute to reducing disparities between rural and urban realities. Population imbalances and intensifying climate impacts are prime challenges for rural areas, which also need to counter diminishing infrastructure and the lack of digital competencies to enhance their attractiveness. Cities, in turn, face their own set of challenges, such as contamination, natural resources exploitation, and high population densities. Local governments have been embracing the Smart City approach to accomplish sustainable development, which might also benefit the revitalization of rural areas if conducted through a tailored regional approach. Enhanced connectivity between rural and urban realities through smartness is, therefore, becoming an important element for the shaping of adaptive, energy-efficient, and resilient communities.

**Keywords:** smart villages; smart cities; rural depopulation; adaptation to climate change; digital transformation; sustainable development

## 1. Introduction

Decades of massive urbanization, combined with spontaneous and unsustainable urban planning efforts, have brought about the deterioration of natural ecosystems around the globe. The chasm between rural depopulation and urban concentration is only expected to widen even further, as it is anticipated that this trend will continue throughout the twenty-first century, an era frequently labelled as the century of cities.

Migration from the countryside to the cities is one of the main obstacles of sustainable development, as it is inducing further pressures on the natural environment. Advancing sustainable development is urgently needed for humanity to be able to mitigate the negative impacts of climate change and safeguard a prosperous future on Earth. The European Union (EU) states that climate impacts are already causing about 12 billion euros of economic losses per year, which could amount to 170 billion euros if global temperatures increase by 3° Celsius from pre-industrial levels [1]. The digital transformation will be an additional hurdle for the shaping of more adaptive, energy-efficient, and resilient communities. Nevertheless, even though smartness is a promising key to accomplishing the United Nations' Sustainable Development Goals (SDGs), one must look far beyond the use of Information and Communications Technology (ICT), as commitments to economic growth and the lack of efficient governance are examples of elements that could hamper the achievement of these objectives [2]. Furthermore, as smart infrastructure can still be vulnerable to climate impacts, the integration of climate resilience into holistic and flexible smart planning tools is of vital importance, so that territorial complexities can be properly governed through a fitting data-driven, pragmatic, and scientific approach [3].

Although current urbanization waves are the most unsettling in developing nations, Eurostat projects that 45.8 percent of the European population will live in predominantly urban regions by 2050, an increase of 24.1 million people compared to 2015, whereas

residents living in intermediate regions (towns and suburbs) and in predominantly rural regions will both decline to 34.2 percent and 20.0 percent, respectively [4]. Furthermore, Eurostat foresees that just over 80 percent of Europeans will live in urban areas by 2050. European integration and urban development assured the evolution of a dense, interdependent, and polycentric network of medium-sized cities on the continent, which are highly heterogeneous but well-connected, even growing into urban conurbations that traverse national borders [5]. Given that the majority of travels are still made over relatively small distances and that most commercial and social activities are still place-based, the distinctive fabric of this network assures that the region is growing as a geographical entity [6]. Since, despite being global issues, solutions to prevent climate impacts such as desertification, extreme weather, sea level rise, and water scarcity would benefit from a specialized regional approach, there is a lot of potential for the region to act as a political organism for strategic decision-making and tailored policymaking.

People migrate to improve their livelihoods, and as the quality of life tends to be higher in urban areas, cities have been a prime destination for many. More economic opportunities, fewer health-related issues, and better access to services and technology are elements that attract people to the cities, despite some of their negative aspects, such as increased crime rates and worse air quality [7]. Governments around the world are progressively placing the concept of Smart City at the core of their governance and policy discourses, not only with the promise of economic growth and enhanced social inclusion but also to make cities more sustainable. Nevertheless, Smart Cities might attract even more migration to the cities, whereas especially the younger generation will be on the move to pursue better education and economic opportunities, which, in turn, would imply a new brain drain from rural areas without proper facilities towards the Smart Cities [8]. On the contrary, the COVID-19 pandemic unleashed radical societal change in almost every dimension of human activity, with social distancing and remote working becoming the standard way of living, which has emphasized the need for digitalization even more. Even in the absence of substantiated demographic conclusions as a result of the pandemic, population density seems to have affected the spread of the virus [9]. Hence, the increased risk of contagion in cities combined with development of remote services and working might even be the catalyst for new migration patterns, notably from the cities to more rural areas.

Climate change, digitalization and migration are, thus, key topics that need to be addressed in sustainable development efforts, not only in metropolitan areas but also in the countryside. The complex dynamics, disparities and territorial realities between rural towns and cities, coupled with current socioeconomic challenges, drive the need for enhanced connectivity between both worlds. For this reason, the term 'Smart Village' has emerged as a solution to address the current imbalances of rural areas. For the benefit of residents and companies, conventional and new networks and services are improved in smart villages using digital and telecommunications technologies, innovations, and better knowledge utilization [10], which is similar to what the idea of a smart city seeks to achieve for cities.

With the above in mind, this article focuses on the question of whether Smart Village proposals could help reduce disparities between rural and urban realities and contribute to the creation of adaptive, energy-efficient, and resilient communities, in conjunction with Smart Cities. The first part of this article addresses the notion of Smart Village, the proposed solutions beyond solely the incorporation of technology and some implemented initiatives, with a special emphasis on the environmental elements. The second part of the article then dives into the rural-urban linkages, such as the role of the Smart City in creating Smart Villages and the much-needed connection at the regional level, as smart and sustainable development practices geared towards preventing climate impacts will require a concerted and holistic approach that is in harmony with the flow of nature.

## 2. Research Aim and Method

According to the European Commission (EC), demographic change and the twin green and digital transition will often affect or accelerate each other [9], as migration exacerbates environmental degradation and natural resource exploitation, while, conversely, it is predicted that biodiversity loss and climate change will influence demographic change. The interactions between these matters are complex but are very important for successful long-term planning and policymaking, as humanity is in for a disaster if the current trends regarding climate change and migration continue. Therefore, the main aim of this research is to understand how the revitalization of rural areas, through the introduction of smart solutions, can contribute to the shaping of adaptive, energy-efficient, and resilient societies. Improved well-being of the villagers, sustainable rural development, and enhanced attractiveness of rural areas are then intertwined with relieving the environmental pressures on cities.

Following an explanatory approach, the purpose of this paper is not to further define the discussed concepts, but rather to generate additional knowledge on the coherence of theories, strategies, and initiatives that could contribute to the academic debate on these matters. However, it is important to outline that there is no universally agreed definition on the term 'Smart City', whilst a common definition of the word 'Smart Village' is even less evident due to its recent emergence. With about 200,000 new Google Scholar articles on Smart City currently published each year, most of the literature explains the notion of Smart City as the deployment of ICT to improve the way of living and working, targeted at cities and communities [11]. One working definition is that "smart cities are all urban settlements that make a conscious effort to capitalize on the new Information and Communications Technology (ICT) landscape in a strategic way, seeking to achieve prosperity, effectiveness and competitiveness on multiple socio-economic levels" [12]. A variety of other definitions are found in the literature [13–15]. Nevertheless, since this urban development model is still very much unexplored, constantly researched, and dependent on societal changes and technological advancement, the definition will constantly be refined as we shape our Smart Cities. An example of this is the incorporation of sustainability into the concept due to the need to prevent climate impacts, leading to the evolution of the notion to that of Smart Sustainable Cities [16].

In addition to the pragmatic definition of Smart Villages by the EC, as outlined in the introduction, the European Network for Rural Development (ENRD) points out that, next to the use of digital technologies, Smart Villages are about people who take ownership of local assets, build new forms of alliances, and think beyond the village itself [17]. By analysing the connection between Smart Cities and Smart Villages, be it through hard (physical) or soft (cooperation and governance) infrastructure networks, we may provide some debate to strengthen the case for the Smart Region as a special entity with decision-making competencies that help reduce vulnerabilities to climate change. Smart Regions are described as heterogeneous urban-rural landscapes that are spatially reframed by digital technology and the corresponding social practices in a range of disciplines (citizenship, governance, economy, environment, mobility, infrastructure), on a discursive, practical, and regulating level. The notion of smart regions emphasizes an integrated approach to the social (re-)construction of smart regions by actors and their networks and adheres to a relational and social constructivist understanding of spaces [18].

With the purpose of scoping this review article, the authors decided to limit their investigation to academic research, policy proposals, and strategic initiatives within the EU, primarily focusing on efforts made in Spain in comparison to other Member States. In addition to the fact that Spain is the country of expertise for the authors, it has also experienced decades of intense depopulation, especially affecting the inner rural areas of the country, for which the government developed a national strategy to combat territorial depopulation [19]. To connect this issue to smartness and the environment, a variety of references have been consulted to bolster the academic foundations of the investigation. Although there is an increasing number of articles that connect Smart

City with sustainability, there is a general lack of literature on Smart Villages and Smart Regions, and even more so on research that specifically connects these concepts to climate change. This limitation is even deepened when explicitly looking for the relation between Smart City and adaptation to climate change. Nevertheless, we were able to find multiple interesting articles to bolster the academic foundations of this paper, using a mix of keyword searches in different combinations, such as 'smart', 'smart cities', 'smart villages', 'adaptation', 'climate', 'environment' and 'sustainability', among others. Smart cities databases such as https://blogs.uoregon.edu/smartcities/resources/ (accessed on 7 March 2022) or "Smart Cities Dive" and other platforms (such as https://www.recursoscientificos.fecyt.es/servicios/acceso-bases-datos, (accessed on 7 March 2022), provided by the Spanish government) contain, among others, relevant and useful information. Another limitation is that most of the literature on these topics has been published very recently. Nevertheless, the authors hope this paper can contribute to future research on this topic, so that researchers can keep connecting these complex matters, through a holistic and transversal modus operandi, in order to make our communities, both in villages and in cities, more resilient to climate impacts.

## 3. Gateway to Smart Sustainable Villages

### 3.1. Depopulation, Climate Vulnerabilities and Digital Competencies in Rural Areas

Many rural areas find themselves in a circle of decline. The trends that mutually reinforce each other include inadequate and diminishing services as well as the absence of sustainable business activities [17]. According to the ENRD, the lack of vital infrastructure and services results in a lower rate of business creation, which makes jobs scarcer. The lack of proper jobs, in turn, assures migration to the cities and population ageing, resulting in lower population density, which prefaces a further decline in critical services. Although rural areas enjoy clearer air, cherish a reduced cost of living, and relish abundant land, the lack of access to vital public services (such as education and healthcare), and private services (such as broadband internet connection) ensure that villages are confronted with their specific set of challenges compared to the cities [9]. Because of territorial depopulation, 48.1 percent of smaller Spanish municipalities lost between 10 and 50 percent of their population in the period between 2001 and 2018 [19]. Similar rates are observed in northern Scandinavia, Germany, eastern Europe, and other southern European nations, and to a lesser extent in areas less clustered around the periphery of larger cities in central and western Europe [9]. Therefore, countering the depopulation of rural Europe is essential to be able to safeguard their vital role in the economy and society, for example in relation to energy and food production, natural resource exploitation, and tourism. To reduce the gaps in development between rural and urban Europe, better public policies and local development plans aiming at restructuring and revitalization processes could increase the growth potential of rural areas [20]. Yet, the rural-urban divide is now more apparent than ever, and people still find country life to be unappealing, particularly the younger generations [21].

Despite being a global problem, environmental impacts caused by climate change such as wildfires, water scarcity, floods, extreme temperatures, desertification, cyclonic windstorms, and coastal erosion occur in relatively small spaces compared to the coarse grids of coupled climate models [22]. According to the policy report *'Climate Change Impacts and Adaptation in Europe'* that provides advice to the EC [23], there is a clear north-south divide with respect to the burden of climate change. Intensifying heat waves and diminishing water resources in southern Europe affect crop yields, energy production, and tree line shifts much more severely than in the north. Hence, climate impacts are rather regional, as they are bound by the natural configurations of the territory. For instance, flood risks will be more common in low-lying zones such as the Netherlands, while conservation of forest and tundra ecosystems will be more relevant for areas with higher woodland density such as in northern and eastern Europe. From this perspective, certain communities are disproportionately and systematically more vulnerable to climate

impacts, whereas rural communities are more often dependent on specific types of natural capital for their business activities and would, therefore, benefit more from community-level adaptation to climate change [24]. Rural communities also tend to lack proper infrastructure and resources to increase their adaptive capacity to natural hazards, which, in turn, could cause environmental migration and affect the supply chain of multiple goods and intangibles to the cities. Therefore, in order to shape a more resilient society, addressing the vulnerabilities of rural communities should follow the socio-economic characteristics and territorial complexities to maximise the benefits from the acquisition of new skills and the ability to absorb innovation [20].

Although new technologies have the potential to induce rural revitalization, especially in relation to overcoming the distance and low population density obstacles, the digital gap between citizens and villagers is still very relevant. According to the ENRD, rural areas are suffering from a triple digital divide, namely, (1) broadband connectivity, as only 47 percent of rural households enjoy Next Generation Access (NGA) compared to 80 percent in the cities, (2) the lack of digital skills, and (3) a lower uptake of digital technologies [17]. The global pandemic has made this digital gap even more visible, whereas technological advancements have rapidly outpaced previous achievements in rural areas and generic policies neglected specific local needs for the extended adoption and use of digital solutions [25]. Even though digital education and the implementation of high-speed digital infrastructure are top investment priorities for stakeholders and rural communities [17], the issue of population ageing remains an important barrier to achieving satisfactory progress. Nevertheless, research from the UK shows that there is indeed a certain willingness of the elderly to adopt digital tools, provided that these technologies support the personal or collective needs of older adults in these rural communities and do not replace alternative modes of delivery [26]. Nevertheless, one drawback is that older people frequently require assistance from younger generations to grasp the vast array of options and configurations that digital appliances support. Therefore, public policy and private initiatives should be focused on insisting on the empowerment of rural populations and the need to prevent resistance to the use of technologies by appealing to these populations' resilience as well as enhancing the allure of rural areas by changing their roles in the digital economy [17,25].

*3.2. Connecting the Smart Village*

As a result of the Cork Declaration 2.0 of September 2016, '*A Better Life in Rural Areas*', the EC launched the EU Action for Smart Villages initiative. This document introduced the concept of Smart Villages to European policymakers with the purpose of empowering local stakeholders through investments in technology, supported by different policy mechanisms such as the Common Agricultural Policy (CAP), the Cohesion Policy (CP), the research and innovation programme Horizon 2020, and the Connecting Europe Facility (CEF) [10]. Subsequently, the Bled Declaration of April 2018 expressed the desire for the concept of Smart Villages to become a model for rural areas in Europe, aimed at redefining rural employment opportunities and enhancing rural participation in the digital economy [27]. The European Parliament (EP), as well as four Directorate-General (DGs) of the EC (AGRI, CNCT, MOVE and REGIO) participated in the Bled Declaration, whereas two more DGs (EAC and ENER) got involved in a later 2018 congress in Gödöllő, Hungary [28]. More investment in smart villages benefits the residents, making their lives more comfortable and simpler; however, it also makes a community more empowered, resilient, autonomous, and connected. It is crucial that smart villages are emphasized on the political agenda in Europe [21].

According to the ENRD, Smart Villages are seen as a convenient solution to (1) respond to depopulation and demographic change, (2) find local solutions to public funding cuts and the centralization of public services, (3) exploit linkages with small towns and cities, (4) maximise the role of rural areas in the transition to a low-carbon, circular economy, and (5) promote the digital transformation of rural areas [17]. Thus, smartness is conjured once

more as the key remedy to resolving the societal challenges of climate change, digitalization, and population disparities. However, it is of critical importance to understand that this panacea consists of many more ingredients than the mere incorporation of technology in rural affairs. Thus, the outcomes should be properly analysed beforehand, so that existing local assets and capabilities can be leveraged optimally, therewith cultivating smart growth. In order to foster this needed synergy between digital tools and human capacity-building, the Bled Declaration formulated eight concepts that would characterize successful frontrunner Smart Villages, namely (1) precision farming, (2) digital platforms, (3) shared economy, (4) circular economy, (5) biobased economy, (6) renewable energy, (7) rural tourism, and (8) social innovation [27]. Efforts to achieve progress in these outlined concepts often go beyond solely the incorporation of ICT and would, for instance, include supportive public policies, intense collaboration between stakeholders, and attitude changes. Regardless, European policymakers believe that rural areas are potential hotspots for new knowledge-intensive workplaces, which would revitalize local industries and grant new opportunities for young adults to pursue their professional specializations beyond the city walls, therefore enabling the transition towards Smart Villages.

The complexity of the numerous dynamics and layers that make up rural life assures that a uniform approach to the creation of Smart Villages is not feasible, which raises barriers against the rapid rollout of smart rural development practices across Europe. From a conceptual perspective, there is still much experimentation and research pending to understand how smart growth policies can be properly tailored to Europe's highly diverse rural areas, with the most isolated regions often being characterized by low accessibility, low education levels, and a negative migratory balance [29]. Evidence from Poland demonstrates that rural areas near large urban agglomerations have better resources and access to new professional technologies, networking institutions, and a closer proximity to expert and management services, which create more favorable conditions for the practical implementation of the smart village concept [20]. Additionally, Scottish research supports the idea that Smart Villages need a community with a high capacity for delivering creative solutions, sufficient digital connectivity, and strong social capital, meaning a strong spatial dimension to their socio-economic performance [30].

From a policy perspective, there is also a need to improve mechanisms that support the transition of rural areas towards Smart Villages, which currently remain complex, fragmented, and therefore not efficiently utilized. The two main funding mechanisms for rural development, namely the Common Agricultural Policy (CAP), and three of the five structural funds of the Cohesion Policy (CP) grant only a small percentage of their budgets to rural development and are especially lacking investments in other facets of the rural economy besides the farming sector [28]. Unfortunately, a recent press statement of the European Committee of the Regions (CoR) and the RUMRA and Smart Villages Intergroup reveals that rural areas are being overlooked in the EU's COVID-19 economic recovery package, Next Generation EU, due to the setup of the structural framework and the allocation of funds through calls for proposals [31]. Thus, in order to safeguard a fairer distribution of funding, there is a need for more engagement with rural stakeholders to involve them properly in the broader decision-making process, so that better technical support and stronger capacity building can be fostered in the European countryside.

### 3.3. Deploying the Smart Village

Digital technologies are believed to be a pervasive and powerful catalyst, able to change societies and economies [2], and therefore, towns and villages that will triumph over the digital transformation will have a solid competitive edge regarding their attractiveness, which is especially relevant for the allure and retention of the younger generation. Improved connectivity as a result of the implementation of ICT solutions in rural areas will help integrate small businesses into e-commerce [25], improve quality of life through communication and eHealth solutions [26], optimize public transportation and smart mobility to boost accessibility [32], and traverse the obstacle of distance by inciting new

opportunities in teleworking and recreation [21]. Furthermore, remote education grants an opportunity for younger villagers to get access to proper tuition [17], while, in a similar vein, advances in eGovernment can bring the public authorities closer to the villagers, improve collaboration networks between different stakeholders (or between towns in a certain region), and facilitate more agile public procedures, such as e-voting. Rural areas can, thus, leverage their existing recreational value, local production specializations, and cultural heritage to address the challenges of depopulation and population ageing using cutting-edge technologies.

In Spain, some important early steps towards the shaping of Smart Villages are already underway. Located in the Pyrenees about 150 km away from the regional capital of Zaragoza, the town of Ansó is participating in the Smart Rural 21 pilot project together with several other European towns, funded by the EC. Thus far, the village of about 405 inhabitants introduced a 100MB optic fibre connection, co-working spaces, an electric car charging point, a housing development and rental assistance plan, and organized community workshops in the local school [33]. Through the participation in the project, Ansó aims to (1) further improve connectivity, (2) facilitate access to housing, (3) encourage entrepreneurship, (4) positively impact the energy transition, and (5) facilitate family and social development. The Mining Area of Cartagena–La Unión is another concrete Spanish initiative that embraces digitalization to incite a transition towards a Smart Tourism destination, connecting their mining and industrial heritage with ICT [34]. Although the technological process was costly and complicated in the beginning, the use of digital technologies made this archaic mining area much more competitive, while facilitating visitors with exclusive content about the landscape, its aesthetic characteristics, and the architectural, environmental, and socioeconomic value of these previously ruinous industrial heritage sites. A report of the Spanish Rural Development Network (REDR) outlines some regional initiatives, such as an Asturian programme for rural entrepreneurs that granted 2.3 million euros to new businesses, and a Catalan energy efficiency advice project that was shared with 100 SME's, 11 public schools, and 47 municipalities [35]. Furthermore, the national government has been actively attempting to improve territorial connectivity in its rural areas, resulting in some concrete actions, such as the deployment of a broadband internet connection in at least 90% of Spanish villages with fewer than 5000 inhabitants, a programme to connect Spanish schools, and a specific programme to develop Smart Rural Territories within the broader National Plan for Smart Territories [19]. The establishment of such policies at different levels of government, vital for technical support and financial resources, will provide communities with the tools needed to start reshaping their local assets through digitalization and social innovation, in order to embed themselves in the digital economy. By the same token, this will enhance the capacity of rural communities to adapt to the impacts of climate change, as "new solutions will help enable a climate-neutral position by changing the paradigm, jumping from standard urban and rural planning patterns to the changed mindset of communities" [36].

The shaping of adaptive, energy-efficient, and resilient Smart Villages is crucial to their long-term sustainability, as is safeguarding progress made in the field of digitalization and population disparities. The mobilization of financial resources, while building on their natural, political and social capital (among others), is vital for adaptation to climate change endeavours [24]. Self-sufficiency and energy diversification, for instance through the utilization of solar power, microgrids, and biomass, will be key for clean and renewable energy provisioning in Smart Villages, whereas these smart energy technologies will also come with additional benefits related to services, skills, health, and employment [37]. Moreover, a self-reliant power supply accelerates the implementation of ICT, while also shielding digital infrastructure from potential climate hazards. Despite the transition to a digital economy, rural communities are still very dependent on economic activities in the primary sector, such as agriculture, forestry, fishery, and mining. Smartness applied in this cornerstone sector of rural life can contribute to faster and more accurate decision-making and more efficient production. Precision farming implies the observation, measurement

and response to inter- and intra-field variability in cultivation, backed by the extraction and analysis of huge amounts of data obtained through ICT [38]. Drones, Internet of Things (IoT) sensors, nanostructured biological sensors, and technologies for smart animal management are some examples of Climate-Smart Agriculture (CSA) techniques used in precision farming that can also help to decrease farming losses, boost yields, and monitor, detect, and possibly prevent plant and animal [39]. By using precise cutting and real-time production and purchase data, for instance, digital forestry solutions help reduce waste, transportation expenses, and unproductive time consumption [38]. Moreover, improved weather forecasting using Big Data and intelligent energy and water systems could make rural communities much more resilient to previously unforeseen climate impacts. On a final note, in a stakeholder reflection for the ENRD, Slee [40] argues that Smart Villages are already covering all elements of the European Green Deal through collaborative action and innovation, with the concept being a very suitable entry point to fulfil the Green Deal's objectives and, thus, offering a novel way to accelerate the green transition.

## 4. Connectivity for Smart and Sustainable Rural-Urban Linkages

### 4.1. Smart City Uplinks for Smart Villages

Smart Villages are born out of the debate on Smart Cities, for which it is important to design public policies based on substantiated experiences, while considering the broader impact at the regional, national, and even global level [41]. Smart City proposals are generally directed at the enhancement of urban services by means of technological, collective, and human capital [12], aimed at cost reduction, increased efficiency, and enhanced interactivity [20], so that life preferences and civil liberties can be exercised more royally and sustainably [41]. Despite the distinctive emphasis on local empowerment in Smart Village initiatives, the replicability of disruptive technologies applied in Smart City pilots, such as 5G, Artificial Intelligence (AI), Big Data, Cloud, and IoT, has the huge potential to enable innovation in rural communities if they are properly adjusted to their needs. After conducting in-depth research on the use of IoT in smart cities and smart villages, Cvar et al. [42] contend that the main differences are caused less by the technology itself and more by the socioeconomic, cultural, and policy specifics of various digital innovation ecosystems, highlighting the differences between different regions and the existence of sizable gaps between the design and implementation of digital transformation.

Cities face a specific set of challenges, which include contamination, energy inefficiency, exploitation of land and natural resources, inefficient urban services, insecurity, and social segregation, among others [3], caused by decades of unsustainable urbanization practices. The continued rush to cities for better economic opportunities and enhanced quality of life will only aggravate these pressures on urban systems, with current migration patterns that only add fuel to the flames. More population in cities will also imply a greater need for energy and natural resources, which come with additional environmental pressures, such as pollution and waste generation. Metropolitan areas are already substantially warmer than their surrounding rural areas due to the urban heat island effect [23]. More frequent extreme weather conditions will affect biodiversity, citizen health, energy consumption, and water and air quality [3], while natural hazards could drastically impact the economic, political, and social stability of affected cities.

To overcome these challenges, city councils have been exploring digital solutions proposed through the Smart City concept for several years now, for example by using ICT as an enabler for high-quality urban services, mixed land uses, and transportation linkages, in search of long-term economic growth [13]. However, the interconnection and synchronization between individual technologies remain major hurdles for cities [16], while the same goes for tight system security. Moreover, like in the concept of Smart Villages, there is no one-size-fits-all solution for the shaping of Smart Cities, as governments at multiple levels have been experimenting with different frameworks, for instance following infrastructure-oriented, geographically based, or economic sector-based strategies [12]. Yet, these different visions and strategic priorities lead to unique ways of integrated development practices

in distinct cities [13], something beneficial to new means of collaboration, exploration and knowledge-sharing. Equally, this underpins the need for more holistic approaches in accordance with the multidimensionality of the concept, whereas a rather technocratic vision of the Smart City has dominated research and strategies since its inception stage, namely as a product of commercial or political interests embodied in isolated efforts [43]. Yet, the social aspects through the participation of people and communities [13,36] and the environmental aspects in favour of enhanced sustainability [3,16], are ever more relevant for shaping inclusive, resilient, and prosperous cities.

Considering that Smart City research is scalable, due to the focus on sustainability and quality of life through the enabling role of ICT, the literature on the Smart City can serve either megacities (macro level), smart cities (mezzo level), or smart villages (micro level), even though they constitute different domains of analysis [41]. For instance, carbon-neutral technologies, automated vehicles, and other smart city projects can all have a good impact on rural affairs [28]. However, there is still much room for improvement because the application of the knowledge gained from Smart City research to rural realities is still in its infancy. This is because the crucial role that the rural–urban linkage has on urban development is frequently overlooked, and because the roles that rural ecosystems play in urban development are not well represented [44].

### 4.2. Reconfiguring Smart and Sustainable Rural-Urban Linkages

Cities tend to be far from self-sufficient, as they are critically dependent on their hinterland for the dissemination of waste and the capitalization of resources, even though this is now more globalized than in the past [16]. But even in a global context, rural areas are still mostly the producers of goods that urban areas consume, while they must deal with the negative externalities of the contaminating output of cities. Rural populations, on the other hand, also benefit from cities through their proximity, for example through their connecting function as a marketplace and their broader offering of goods and services [6]. Despite this vital interdependence between villages and cities, "rural regions do not have the same access to resources and markets and differ in terms of socio-economic conditions and social structures" [29]. Cities, on the other hand, require the integration of their hinterland to expand their size of functional reach, and therewith enhance their economic growth and competitiveness [6]. Nonetheless, urban and rural areas are still regarded as separate entities in most planning processes [18]. When rethinking rural-urban linkages, efforts should be geared to the development of inclusive and participatory networks, which are grounded in cooperation instead of hierarchy, so that the whole region benefits from both the digital transformation and the green transition. Figure 1 below offers an oversimplified but imaginable illustration of such smart and sustainable linkages.

Enhanced connectedness bridges the gap of physical distance between rural and urban realities, while social unrest, the COVID-19 pandemic, and environmental issues in cities open the door to new ways of working and living, benefitting the attractiveness of villages. Advances in sustainable mobility and flexible transport systems, generally a major focus in Smart City frameworks, are very promising for rural areas, especially in terms of cost-effectiveness, efficiency, and performance of transportation services [32]. Disruptive innovation in the field of Smart Mobility also permits easier teleworking and daily long-distance commuting, boosting overall accessibility and facilitating remote information sharing [6]. Despite the need for simplification in investment and planning mechanisms combined with better coordination between authorities, almost all technological solutions in mobility can be applied to both rural and urban contexts, which have the huge potential to foster economic growth and social inclusion in the countryside [32]. Better access to villages will reduce depopulation and promote the inflow of visitors, tourists, and possible new inhabitants that bring vital human capital with them.

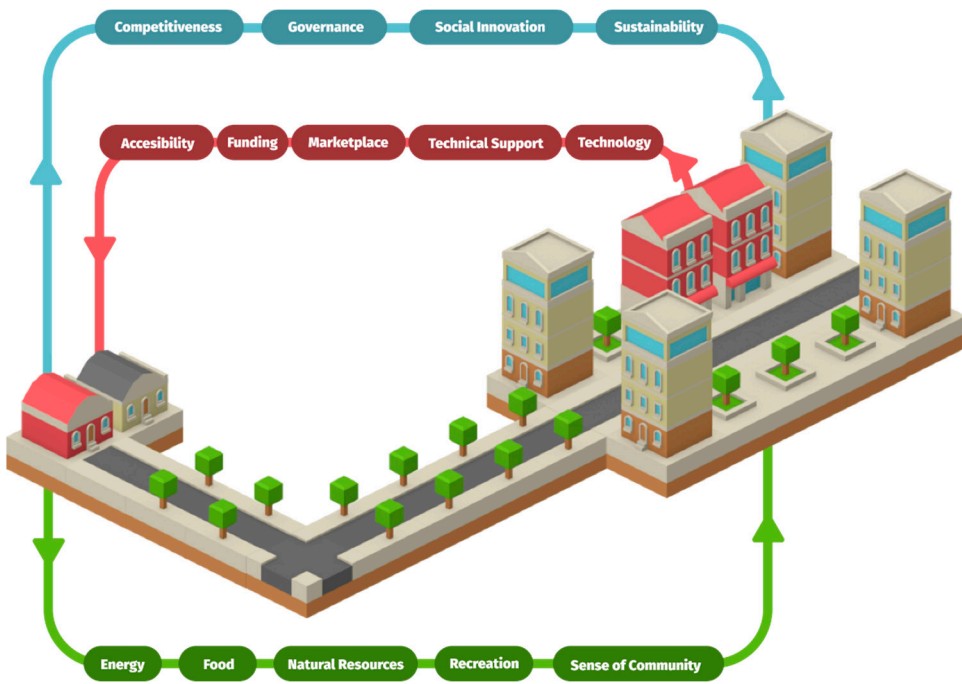

**Figure 1.** Smart and Sustainable Rural-Urban Linkages (Own elaboration).

Investment in hard infrastructure, such as transit systems and digital networks, by itself will not be enough incite effective smart regional development, as opportunities enabled by these fresh assets are to be leveraged by people and their communities. Soft capabilities related to governance and collaboration between stakeholders will be the differentiating factor, although topics like socio-cultural urban-rural relations, participatory schemes, and behavioural facets are still barely considered in the shaping of Smart Regions [18]. The sense of belonging together is far more relevant in rural areas than it is in cities, and therefore, considered vital for the effectiveness of the proposed bottom-up approaches and community-centred development in Smart Villages [36]. In the regional context, these feelings of solidarity can be a fundamental component that villagers can transfer to communities in cities, so that Smart City visions correspond to their needs and therefore improve the citizen's overall wellbeing and satisfaction. Thus, significantly more emphasis should be placed on participatory processes in smart development efforts, not only to facilitate close collaboration networks among social actors but also to promote co-creation as well as heterogenous and decentralized forms of governance, aimed at reducing disparities [18]. However, progress in reducing population disparities and digitalization efforts should be sustainable long-term, that is, safeguarded from climate impacts.

Climate models used to enlighten policymakers would benefit from stronger and more dynamic downscaling through the elaboration of more refined regional simulations, which, in turn, would contribute to the optimization of adaptation efforts [22]. Enhanced climate predictions powered by the processing of big data, for example, obtained from IoT sensors, can mitigate vulnerabilities to natural hazards in the region, protect natural ecosystems, and optimize land use. Ecosystem services, which are the benefits humanity gains from natural processes in the environment, are especially relevant when reconsidering traditional rural–urban linkages, as they have wider spatial dimensions and are considered a critical link between both worlds [44]. For instance, different types of land such as woods, waterways, and cropland offer vital services to communities, such as outdoor recreation, potable water, flood prevention, and moderation of the climate in human settlements [6]. Distinct types of ecosystem services, such as provisioning services (food, water, and natural resources supply), regulating services (air quality, soil fertility, and waste treatment), supporting services (nutrient cycling and biodiversity), and cultural services (tourism, spiritual essence, and aesthetic value), further emphasize the interdependence between villagers

and citizens [44]. On the other hand, digitalization in agriculture promises to increase overall yield and quality while reducing environmental pressures in addition to facilitating new business models, streamlining the supply chain, and granting new opportunities for collaboration between farmers, for example through the sharing of data [38]. Lastly, renewable energy and its distribution through smart grids throughout the region will accelerate deployment of ICT solutions and safeguard them from climate impacts, thus empowering and sustaining communities in the long-term [37]. All in all, environmental sustainability and social innovation, enabled by the digital transformation, will redefine the role of rural areas in Europe's newly forged digital economy and, therewith, reconfigure spatial disparities through the shaping of Smart Regions.

*4.3. Smart and Sustainable Regions in Spain*

The effectiveness of territorial governance, sustainable transport expansion, or natural resource regulation is intrinsically linked to understanding the full regional narrative and is rather constraining if limited to a specific setting such as a building, a neighborhood, or even a city [43]. Therefore, the needed investments to foster the digital transformation in the countryside, including the uptake of digital skills, would benefit from a strong regional policy framework. According to Matern, Binder, and Noack [18], one can refer to the idea of a smart region when three essential components—discourse, implementation, and regulation—are attainable on a regional level. These authors note that in the case of the Helsinki Smart Region, the implementation dimension is represented by pilot programs for driverless electric buses, examples of on-demand public transportation, initiatives aimed at transforming socially excluded groups into digitally active citizens, or co-working spaces [18]. Regulation, on the other hand, is concerned with the presence of regulatory systems and their connections to regional, national, and European policies as well as financing sources.

In Spain, the National Plan for Smart Territories, published in 2017 and derived from results and experiences from the preceding National Plan for Smart Cities (2015–2017), already changes the Spanish discourse on smart development from a city-level approach to a broader territorial perspective. The plan proposes three specific lines of action, namely (1) Territorial Actions, which consist of the six specific focus areas of Smart Tourism, Internal City Objects (relating to buildings, stations, port and airports), Virtual Interoperability Laboratory, 5G, Smart Rural Territories, and Public Services 4.0 on city and rural platforms, (2) Support Actions that relate to facilitating efforts in the areas of diffusion, governance and standardization, and (3) Complementary Actions, which focuse specifically on the wider topics of IoT for public services and Smart Mobility [45]. Co-financed by the European Regional Development Fund (ERDF), the first calls for proposals through this plan were in the areas of Smart Tourism (25 funded projects with a total investment of 73.97 million euros), and Internal City Objects (eight funded projects with a total investment of 32 million euros) [46]. Nevertheless, most of the grants provided through these calls were allocated to sizable cities or municipalities that already dispose of a certain inflow of tourists, thus having a relatively low impact on the population disparities in the predominantly rural communities of Spain.

Regional initiatives for the shaping of Smart Regions also exist in Spain. One example is AndalucíaSmart, a strategic plan to stimulate smart development in Spain's largest autonomous community in terms of inhabitants, of which a significant proportion are rural communities. In the Action Plan AndalucíaSmart 2020, the regional government is collaborating with the ERDF to promote smartness in seven strategic areas, namely (1) Governance, (2) Finance, (3) Security, (4) Education, (5) Legal, (6) Technology, and (7) Infrastructure, across different sectors, such as tourism, core capital, and sustainability [47]. Although there is still a huge focus on the creation of Smart Cities, the strategy also underpins the inclusion of smaller municipalities. The plan contains twelve initiatives with allocated budgets and performance indicators, of which many will also affect rural communities, such as the establishment of a network of stakeholders, a portal to stimulate

collaboration among municipalities, and creation of different research laboratories. Other Spanish autonomous communities are also laying the foundations to transition towards Smart Regions. Two examples are the Basque Country, which is supported by academics and the municipal collaboration network EUDEL [48], and Cantabria, which is taking important experiences from the Smart City strategy of its capital Santander to stimulate the Cantabrian Smart Region, including virtual reality applications, new payment methods and drone technology [49].

Zooming in specifically on the proposed creation of Smart Rural Territories in the National Plan for Smart Territories, the Spanish government highlights the need for sustainable rural development to counter rural depopulation, and in doing so, diminish rural services and social cohesion fractures [45]. As part of their intervention proposition, five requirements are presented, namely (1) adequate characterization of rural municipalities, (2) identification of territorial provision spaces, (3) attainment of equality in the provision of rural services, (4) institutional strengthening to support the process of rural economic development, and (5) development of technical recommendations. A budget of 51 million euros has been allocated for this initiative and will start with normalization efforts, pilots, and calls for proposals, congresses, and further research. Meanwhile, different calls for proposals are launched at the regional level, such as the 'Smart Villages, innovative solutions for sustainable villages' initiative launched by the Government of Galicia, which started on the 23rd of November, 2021, and already had 168 submitted proposals by December 25th [50], as well as a call for proposals in April 2021 from the Government of Extremadura, consisting of a budget of 2.5 million euros to promote smart development in villages of less than 20,000 inhabitants [51]. Lastly, a private sector project, driven by the Cantabrian newspaper El Diario Montañés, ZWIT Project and Next International Business School (https://territoriorruralinteligente.es/) (accessed on 7 March 2022), was launched in 2020 to provide professional expertise on the development of Smart Rural Territories, supported by 25 partners and collaborators, including Banco Santander and the Cantabrian Chamber of Commerce. All these efforts, among various other existing and future initiatives, imply that the transition from city-centred smart development towards the shaping of Smart Regions is well underway in Spain, which will incite exploration and experimentation in days to come.

## 5. Discussion and Results

Despite their being included in European policymaking only recently, Smart Village initiatives are well underway, even though most projects are still very much experimental and dispersed. To exceed all expectations regarding the improvement of quality of life in the countryside, there is a need for more emphasis on breaking the circle of decline in rural areas. Local empowerment, the implementation of technology, access to funding, the simplification of policy mechanisms, and the continuous strive for experimentation and research all have important roles in diffusing the Smart Village concept. There is an academic consensus that policies should be based on social innovation, a holistic approach, and bottom-up community-led action [20,21,26,30,36]. Nevertheless, "smart development is not a one-size-fits-all concept and its application in rural contexts needs to be combined with a place-based approach adjusted to fit the specifics of rural contexts and linkages" [29]. This strengthens the case for tailored regional approaches, founded in intensive collaboration networks for increased capacity building and technical support. The absence of different forms of capital in rural areas needed to leverage smart rural development can be found in nearby cities, which will be amplified by advances made within their Smart City frameworks, as these experiences and knowledge can be transferred to rural affairs [41]. However, this also implies that the frontiers between urban and rural realities need to be redefined and analysed through a more profound hybrid perspective, as centralization and peripheralization will only accelerate spatial polarization due to the different means and velocities of digitalization [18]. Therefore, the opportunities provided by ICT should be (re)considered and included in strengthening the collaboration between (all levels of)

decision-making authorities, communities, developers, and researchers as well as envisioned in a way that enhances links between rural and urban areas in order to successfully develop a Smart Region under the auspices of a cooperative network of Smart Cities and Villages [36].

Through digitalization efforts, such as the deployment of physical ICT infrastructure, sustainable transportation networks, and the utilization of softer online collaboration tools, citizens and villagers will be able to relish a closer connection, which will help reduce disparities between both worlds. Enhancing the attractiveness of rural life is a core element needed to counter depopulation in rural areas. New opportunities arise with more flexible ways of working, the introduction of knowledge-intensive job opportunities, and new ways to expand the reach of existing assets in the digital economy, for instance by leveraging product specializations or cultural heritage. Such efforts provide an answer to population disparities and digital skill uptake, but progress in these fields should be safeguarded from climate change in order to be sustainable long-term. Thus, when shaping smart and sustainable regional networks, the spatial boundaries of the natural world should be taken more into account, even if they might traverse municipal borders, so that ecosystem linkages can be leveraged to their fullest in the transition towards adaptive, energy-efficient, and resilient communities. For instance, we can imagine a collaboration network between towns and cities critically depending on a certain water body to safeguard a sustainable water supply. Nevertheless, as the battle against climate change requires an immense common effort and possibly even cultural change, opportunities should not only be explored in hybrid rural–urban linkages but also in their separate realities. For instance, green roofs and constructed wetlands in cities can provide new approaches to supporting biodiversity, which would be less feasible in villages [52], while increased optimization of agricultural yield would be less evident in cities.

Reflecting on the theory behind the concept of Smart Villages certainly grants a new perspective and positive outlook as a possible answer to rural depopulation and the mitigation of climate impacts, especially when taking current and future solutions provided by the digital transformation into consideration. A vital component of encouraging the digitalization of rural areas is through proper policy design, and it is exactly here where there is still much room for improvement, especially when looking at the integration of Smart Villages in European policies and their supportive mechanisms. However, there is also a need for more simplification, better communication, enhanced innovation, and ensuring that public policies beyond rural development are aligned with the needs and realities of rural communities [28]. Despite the clear direction towards the creation of Smart Villages through the Smart Region, our analysis of different efforts in Spain unveils these fragmented initiatives, especially on the regulatory side. A conscious effort should be made to shape a clear structure and alignment between different public policies and strategies at various levels of government, so that rural areas are provided with sufficient capacity to unleash social innovation. Future research and practical experimentation are vital for the creation of smart and sustainable networks that connect urban and rural realities, so that disparities between them may be reduced, which the authors hope to incite through the present manuscript, in order to keep moving forward and creating adaptive, energy-efficient, and resilient communities on a prosperous planet.

## 6. Conclusions

Smart Villages must be attractive, competitive and sustainable to break their circle of decline and successfully adapt to the challenges presented by climate change, depopulation, and digitalization. Progress in these areas, through the deployment of technological solutions that enable innovation in rural communities, will redefine the role of rural areas in Europe's newly forged digital economy. As our ways of living and working are increasingly becoming more digital and flexible, enhanced attractiveness of villages can help to alleviate cities on matters such as population density and environmental pollution. At the same time, more capital and information in rural areas will pave the way for further

innovation and a sustainable supply of goods and natural resources to the cities. This, however, will require much more experimentation and exploration, of which the initial return on investment might not be as clear in every specific case, especially when compared to outcomes of disruptive technologies in cities. Therefore, there is a crucial need to leverage the Smart Region to enhance connectedness between citizens and villages, so that knowledge and resources can be optimally leveraged to facilitate sustainable development in the whole region.

This research has analysed the proposition of the concept of Smart Villages as well as the extent to which it can provide an answer to reducing disparities between urban and rural realities, while contributing to the shaping of adaptive, energy-efficient, and resilient communities. We have pointed out that the theory provides a diverse set of opportunities in different sectors of the rural economy, enabled by technological and social innovation, indeed showing mighty potential to facilitate the twin digital and green transition. Nevertheless, on one hand, there is still a huge absence of practical experiences that will define the essential elements that make up a successful Smart Village. On the other hand, there is also a need for well-structured processes regarding policy design and access to funding, which should be concerted holistically between different levels of government. Therefore, the Smart Region and the knowledge transfer from experiences in the field of Smart City should be leveraged to facilitate smart growth and sustainable development in rural areas. Examples from Spain indeed show that efforts are being made to achieve this, implying that future actions will define the Smart and Sustainable Region. Constant research and innovation will help to fine-tune this process.

On a final note, and in order to shape a concerted and holistic approach that will lead to the creation of adaptive, energy-efficient, and resilient hybrid urban-rural communities, one may take inspiration from nature. The Mycorrhizal Network, or Wood Wide Web, is an extensive underground web of fungi and roots that exists below natural spaces, which connects different species of plants and trees. This interconnected web of organisms grants the transfer of nutrients between the flora and the soil to benefit the whole ecosystem. Nature, thus, displays itself as a prime example that can serve as a metaphor for our smart and sustainable networks.

**Author Contributions:** Conceptualization, C.G.F. and D.P.; methodology, C.G.F. and D.P.; investigation, C.G.F. and D.P.; writing—original draft preparation, D.P.; writing—review and editing, C.G.F. and D.P.; corresponding author, C.G.F. All authors have read and agreed to the published version of the manuscript.

**Funding:** This research received no external funding.

**Data Availability Statement:** Data sharing not applicable.

**Conflicts of Interest:** The authors declare no conflict of interest.

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
