# Peer review of "Connecting the Smart Village: A Switch towards Smart and Sustainable Rural-Urban Linkages in Spain"

_land, doi:10.3390/land12040822_

Round 1

Reviewer 1 Report

Thank the authors for the submission of the manuscript entitled “Connecting the Smart Village: A Switch towards Smart and Sustainable Rural-Urban Linkages in Spain”. The study aimed to understand how smart development in the countryside could help reduce disparities between rural and urban realities and contribute to the development of adaptive, energy-efficient, and sustainable communities. The authors clearly introduced the objectives of this study and revealed implications for decision-makers based on their research findings. Although some novel ideas are presented, I have some comments for the authors to improve the manuscript.

(1) In the submitted PDF version of the manuscript, all the headings in the body text start with “1”. Please correct that.

(2) In the reference section, the font style is not consistent across papers cited. Please correct that.

(3) Please add line numbers to the body text of the manuscript so that when reviewers make comments to the authors it is easy to find the issue and solve it.

(3) It might be more proper to use 'this study' or 'this paper' instead of 'this manuscript' in the body text (e.g., in the second paragraph of the 4th page)

(4) In the first paragraph of the 5th page, regarding the effects of climate change (increased frequency of severe weather conditions), could you please also use some numbers from other studies to show the extent to which it affects rural communities?

(5) Rather than directly citing the original sentences from other studies, could you please try to transfer some of those original sentences to statements that can support what you want to express?  

Reviewer 2 Report

The manuscript is really comprehensive and well written. Just a few concerns I would be happy if the author could address.

First is the formatting of the headings, and sub-headings. This style of of 1. for all main headings and 1.1 for all sub-headings is new to me. Therefore, if it is not the prescribed format by the journal, I advise this be revised.

On the methodology. I find the description of the data collection method rather scanty.  Being a review article, what sort of review methodology did you use, and how was this actually carried out to arrive at the used materials, and articles. Consider revising the methodology section to improve it for the purpose of replication.

In the "Smart and Sustainable Cities in Spain" cases provided, Although you highlighted some steps taken in this cases, I was looking to see you discuss to some extent how these connect to directly or relate to the mitigation of climate impacts, or if there are any data on in that direction. Therefore, if you could me a bit further in this regard, it will be great.

Finally, I understand this paper focuses on the European context and more specifically Spain. However, I am just being inquisitive, and something that will also help widen the papers readership and impact if considered as the journal has a broader audience.  How, or in which ways do you consider, based on your experiences with the concept, that your audience, particularly from developing countries can adapt this in their contexts where for instance funds, and other requisites  seen in this paper might not be readily available.

Thank you, and I look forward to your ideas on these few comments   

Round 2

Reviewer 1 Report

Thank the authors for resubmitting the manuscript.

They did a good job revising it.

I have no other comments.